# Current Status of Oral Disease-Modifying Treatment Effects on Cognitive Outcomes in Multiple Sclerosis: A Scoping Review

**DOI:** 10.3390/bioengineering10070848

**Published:** 2023-07-18

**Authors:** Vincenzo Carlomagno, Massimiliano Mirabella, Matteo Lucchini

**Affiliations:** 1Fondazione Policlinico Universitario Agostino Gemelli IRCCS, UOC Neurologia, 00168 Rome, Italy; 2Department of Neuroscience, Università Cattolica del Sacro Cuore, Centro di ricerca Sclerosi Multipla (CERSM), 00168 Rome, Italy

**Keywords:** multiple sclerosis, cognitive impairment, fingolimod, siponimod, ozanimod, cladribine, dimethyl fumarate, teriflunomide, symbol digit modality test, SDMT, paced auditory serial addition task, PASAT, Brief International Cognitive Assessment for MS, BICAMS, neuropsychological assessment, magnetic resonance imaging (MRI)

## Abstract

Introduction. Cognitive impairment represents one of the most hidden and disabling clinical aspects of multiple sclerosis (MS). In this regard, the major challenges are represented by the need for a comprehensive and standardised cognitive evaluation of each patient, both at disease onset and during follow-up, and by the lack of clear-cut data on the effects of treatments. In the present review, we summarize the current evidence on the effects of the available oral disease-modifying treatments (DMTs) on cognitive outcome measures. Materials and Methods. In this systematised review, we extract all the studies that reported longitudinally acquired cognitive outcome data on oral DMTs in MS patients. Results. We found 29 studies that evaluated at least one oral DMT, including observational studies, randomised controlled trials, and their extension studies. Most of the studies (*n* = 20) evaluated sphingosine-1-phosphate (S1P) modulators, while we found seven studies on dimethyl fumarate, six on teriflunomide, and one on cladribine. The most frequently used cognitive outcome measures were SDMT and PASAT. Most of the studies reported substantial stability or mild improvement in cognitive outcomes in a short-time follow-up (duration of most studies ≤2 years). A few studies also reported MRI measures of brain atrophy. Conclusion. Cognitive outcomes were evaluated only in a minority of prospective studies on oral DMTs in MS patients with variable findings. More solid and numerous data are present for the S1P modulators. A standardised cognitive evaluation remains a yet unmet need to better clarify the possible positive effect of oral DMTs on cognition.

## 1. Introduction

Multiple sclerosis (MS) is a chronic inflammatory demyelinating disease of the central nervous system, characterised by progressive neurodegeneration with neuroaxonal damage leading to pronounced brain matter atrophy [1,2]. Cognitive impairment (CI) is one of the clinical features of MS with a variable prevalence in the adult MS population ranging between 34% and 65%. CI is also observed in paediatric MS, in clinically isolated syndrome (CIS), and in radiologically isolated syndrome (RIS) [1,2,3]. These findings highlight the importance of CI as an early indicator of disease activity that sometimes may anticipate the classical clinical manifestations [4,5,6]. Longitudinal studies have shown that CI progresses over time in people with MS [7,8], and it has been observed that cognitive disorders have a higher prevalence and severity in progressive MS [9,10]. Not all patients show the same rate of progression of CI. Age at onset (younger or older), higher lesion load on a brain MRI, and higher rate of brain atrophy within the first two years after MS diagnosis represent some of the worst prognostic factors for CI [7,11,12,13].

Brain atrophy is a common MRI finding in MS patients and has been associated with cognitive decline [14]. Data from several studies show that grey matter bulk is indicative of cognitive integrity; in particular, the volumetry of the thalamus, neocortex, and mesial temporal cortex correlates with cognitive performance [15,16,17]. Furthermore, the advent of new MRI imaging techniques, such as double inversion recovery (DIR) sequences, have made it possible to identify cortical lesions, whose presence is associated with CI [18].

The most frequently involved cognitive domains are information processing speed and memory [19,20]. However, other cognitive domains, including executive functions, verbal fluency, complex attention, visuospatial perception, and social cognition can be significantly involved [3,20,21,22,23].

The cognitive hallmark of MS is represented by an impairment of information processing speed, which is found from the very first clinical and even pre-clinical stages of the disease [4,6,10,19,20,24,25]. Information processing speed is fundamental to higher cognitive function, and its impairment can also impact other cognitive domains [24].

An accurate management of cognitive disorders in MS includes early screening and timely follow-up during the disease course and a series of interventions and compensation strategies to optimise the patient’s global functioning and their implementation in daily activities.

Numerous disease-modifying treatments (DMTs) are currently available, and they have dramatically changed the natural history of MS. These therapies have a significant impact on annualised relapse rate (ARR), expanded disability status scale (EDSS) worsening, and radiological activity [26]. By contrast, there is less evidence of DMT’s effect on CI. Over the last decade, the range of MS therapies has expanded considerably thanks to the introduction of oral DMTs, allowing the patient a more pleasant route of administration and increasing the clinician’s power to tailor therapy to the individual patient. Moreover, oral DMTs seem to have neuroprotective function as seen in preclinical studies [27,28,29,30]. Data from the literature also suggest that the process of brain atrophy appears to be slowed down in patients treated with DMTs, including oral drugs such as the S1P modulators [31,32,33] teriflunomide [34] and dimethyl fumarate [35,36]. Some studies have shown a beneficial effect of oral DMTs on cognitive functioning, but to date, data are still limited [37,38].

Recently, the focus on the impact of oral DMTs on CI has considerably grown, also considering that the recent RCT on ozanimod (SUNBEAM) included the symbol digit modalities test (SDMT) as secondary outcome [39].

The goal of this systematised review is to analyse, according to the current scientific literature, the potential effect of oral DMTs on CI in MS.

## 2. Methods and Search Strategy

In this systematised review, all the studies that longitudinally assessed CI in MS patients treated with an oral DMT were extracted. To perform a systematised literature review, we conducted electronic searches using MEDLINE, EMBASE, and Web of Science Core Collection from database inception to 28 July 2022. After removing duplicate studies, we detected 818 articles. Two independent screeners, VC and ML, examined the references and extracted the data regarding DMTs and the reported cognitive outcome measures in each study. Exclusion criteria were the following: paper not concerning cognition; animal studies; paper not concerning oral DMTs; reviews; case reports; book chapters; editorials; cross-sectional studies. References from these articles were also manually searched by the screeners to be more inclusive. Descriptive statistics were represented as median and range.

## 3. Results

The keywords used for the search were (multiple sclerosis) AND (fingolimod OR ozanimod OR siponimod OR teriflunomide OR cladribine OR dimethyl fumarate) AND (cognit*). Out of the 818 detected articles, 715 were excluded for ineligibility (not concerning cognition; animal studies; not concerning oral DMTs; reviews; case reports; book chapters; editorials). 103 articles were then analysed, of which 75 were excluded because they did not provide sufficient data on cognitive assessments. Therefore 29 studies were included in this review (Figure 1).

We found 29 studies that longitudinally evaluated CI in MS patients treated with an oral DMT. The vast majority (*n* = 20) provided data on S1P modulators [fingolimod (FTY), ozanimod (OZN) and siponimod (SIP)], while we found six studies with data on teriflunomide (TFL), seven on dimethyl fumarate (DMF), and one on cladribine (CLAD). Most of the studies (*n* = 25) evaluated a single oral DMT, while three papers included patients taking two different oral DMTs (1 FTY and DMF; 1 FTY and TRF; 1 FTY and SIP) and one evaluated patients on FTY, DMF, TFL, and CLAD.

### 3.1. Sphingosinte-1-Phosphate Modulators

The majority of studies providing data on S1P modulators concerned FTY (*n* = 16), while a minority analysed OZN (*n* = 2) and SIP (*n* = 2), with one of these studies collecting data on both FTY and SIP (Table 1). The reviewed papers came from observational prospective studies (*n* = 13), randomised clinical trials, and their extension studies (*n* = 7) with a median duration of 21 months (range 6–120). The total number of patients was 4935 (2765 on FTY, 1272 on SIP, and 898 on OZN), of which 3360 had relapsing remitting MS (RRMS), 303 had primary progressive MS (PPMS), and 1272 had secondary progressive MS (SPMS). The median number of enrolled patients was 84 with a range of (8–1575). The most frequently used cognitive assessment tools were the SDMT (in five papers as the sole assessment tool, and in seven together with other tests) and the PASAT (alone in four papers, and in combination with other tests in 11).

In 16 studies (80%), there was a comparator, which was interferon in 7 studies (35%), placebo in 5 (25%), while in one study a pool of healthy controls was used as a comparator [55].

In 14 studies (70%), patients treated with S1P modulators showed a significant improvement in cognitive performance over time. Three of these studies [40,49,53] compared S1P modulators with placebo and reported a significant difference between groups, with S1P modulator-treated patients performing better than placebo. Four studies reported a better performance of S1P modulators compared to an injectable DMT (two studies with IFN; one with GA; and one with both IFN and GA) [39,43,46,50]. Other DMTs were used as comparisons such as Natalizumab (NTZ) (three studies) [41,47,56], Alemtuzumab (ATZ) (one study) [56], DMF (two studies) [56,57], and TFL (two studies) [45,56]. In these studies, no significant differences were found with these DMTs, with the exception of the study by Glasmacher et al. [56] where TFL and DMF were associated with significant worsening over time in cognitive assessment scores compared to FTY and CLAD.

MRI data on brain volume loss (BVL) were collected in 12 studies. Compared to placebo, S1P modulator treatment demonstrated a significant reduction in BVL in all of these studies. Regarding a direct comparison with other DMTs, three papers [39,42,43] found a lower BVL in the S1P modulator group compared to IFN and/or GA, while two other comparative studies vs. GA and NTZ, respectively [46,47], did not find any significant difference.

Moreover, two papers found that a lower BVL during S1P modulator treatment correlated with better cognitive test scores [53,55]. Two studies were mainly focused on thalamic atrophy, demonstrating that patients treated with S1P modulators have a smaller reduction in thalamic volume compared to placebo [33], which could also be predictive of CI [55].

Only two studies include data on neurofilaments (sNFL) [54,57]. In one of these studies, higher levels of sNFLs correlated with greater CI and BVL, and patients treated with S1P modulators had a significant reduction in sNFL levels compared to placebo [54].

### 3.2. Dimethyl Fumarate (DMF)

Seven different studies evaluated clinical cognitive outcomes in DMF-exposed patients (Table 2).

These studies included six prospective studies and one study reporting the pooled data from RCTs and their extension studies. The overall median study duration was 12 months (range 12–24). The total number of DMF patients was 1768, with a median of 148 (range 16–769) for each study. Following DMF clinical indication, all the included patients suffered from RRMS. PASAT was used as an assessment tool in most studies (*n* = 4) [56,58,60,62], and the SDMT, the Processing Speed Test (PST) and the Rao’s Brief Repeatable Battery (BRB) with Stroop Test were also used in four studies [57,59,61,62], respectively.

Three studies included only patients exposed to DMF, while two studies also included other DMTs. The remaining two studies had a placebo control group and a healthy subject’s control cohort.

Observation of the data shows that in four studies, patients presented a longitudinal improvement in cognitive performance over time [58,60,61,62], while in two studies, cognitive assessment remained stable with no significant differences vs. FTY [57,59]. On the contrary, there was a significant worsening of PASAT in the study by Glasmacher et al., where DMF was analysed together with TFL as Category 3 vs. Category 1 (NTZ and ATZ) and Category 2 (FTY and CLAD) DMTs [56].

MRI data on BVL came from only one prospective study without comparator where there was a percentage reduction in brain volume from a baseline of 0.12% at 6 months and 0.24% at 1 year [62]. One study also collected data on sNFLs, showing higher levels in FTY-treated patients. sNFL levels were assessed with an average timeframe of 7.1 months from the first study enrolment visit, and there was no follow-up data to assess the trend of sNFLs over time [57].

### 3.3. Teriflunomide

Six studies evaluated clinical cognitive outcomes in MS patients during TFL treatment.

Most of the papers included were observational prospective studies (*n* = 5), but there was also a post hoc analysis of the RCT TEMSO and its extension [63] with a median duration of 24 months (range 12–58) (Table 3). There were a total of 1020 patients, with a median of 88 (range 4–594). All enrolled patients were diagnosed as RRMS. SDMT was the only cognitive assessment in two studies, while the other four used PASAT (*n* = 2), Brief International Cognitive Assessment for MS (BICAMS) (*n* = 1), and Rao’s BRB (*n* = 1).

Two studies evaluated patients on TFL treatment longitudinally without any comparator group. Coyle et al. reported substantial stability in the SDMT score during a two-year follow-up in a large cohort of patients, while Bencsik et al. found a mild improvement in BICAMS at 12- and 24-month evaluations [64,66].

The remaining four studies also evaluated patients who did not undertake TFL. One study included a healthy subject control group [65], while another included a placebo control group [63]. The last two studies included patients on other DMTs [45,56].

In the TEMSO RCT and its extension study, a mild improvement in PASAT Z score was found over time. Moreover, patients who first started TFL in the core phase of the trial better performed in PASAT 3 than patients who switched to TFL in the extension phase [63].

In their study, Corallo et al. found that the Rao’s BRB scores remained stable over time in TFL-treated patients [65].

Regarding the potential comparison with other DMTs, Glasmacher et al. reported that TFL and DMF treatment was associated with a significant worsening of the PASAT score over time compared to the higher efficacy DMTs [56], whereas in the prospective study by Guevara et al., 8/45 patients of the full cohort (only four patients in TFL cohort) had a SDMT that decreased 4 points or more at year two [45]. However, TFL sample size was very low in both studies, and the reported data do not separate individual treatments.

An analysis of BVL was performed in three studies [45,63,65]. In the study by Guevara et al., a significant reduction from baseline in white matter (WM) volume, peripheral gray matter (GM), and whole brain volume was observed, without significant inter-group (different DMTs) differences [45]. Another study showed a mild increase in GM volume compared to the baseline in TFL-treated patients [65], whereas in the TEMSO study and its extension, lower BVL on TFL treatment correlated with higher PASAT-3 scores [63].

### 3.4. Cladribine

The data on Cladribine (CLAD) came from a single study. Three categories of DMTs were distinguished in the study design, where Category 1 included NTZ and ATZ, Category 2 included FTY and CLAD, and Category 3 included DMF and TFL. The total number of study subjects treated with CLAD was 13, and analyses of the results were performed per category of DMT. Category 2 (FTY and CLAD) was associated with significant improvement in the PASAT scores, while Category 3 was associated with significant worsening. Stability in the scores was seen in those treated with Category 1 DMTs [56].

## 4. Discussion

The focus on the non-motor symptoms of MS has grown considerably in recent years, as there has been an increased awareness of their impact on the daily-living activities of people with MS (PwMS). Certainly, cognitive disorders represent one of the most difficult challenges for patients and clinicians. The assessment of cognitive disorders in MS comprises several reliable and valid assessment tools. Among these, the Minimal Assessment of Cognitive Function in MS (MACFIMS) allows for a complete neuropsychological assessment of cognitive disorders associated with MS, although it has the disadvantage of a more time-consuming administration [67]. Other more rapidly used instruments, such as the Brief Repeatable Neuropsychological Battery (BRNB) [3], the Brief International Cognitive Assessment in Multiple Sclerosis (BICAMS) [68], and the Symbol Digit Modalities Test (SDMT) alone, on the other hand, do not allow for a complete neuropsychological assessment but can evaluate the cognitive functions most frequently affected in MS. BICAMS and SDMT have the shortest administration durations, offering an excellent balance between duration and clinical validity. Cognitive assessments with validated tests should be performed at baseline and once a year thereafter with the same assessment tool in all patients with clinical and radiological evidence consistent with MS [69].

Treatment options for CI in people with MS include cognitive remediation, cognitive exercise, and pharmacological treatments. Recent evidence emphasises the beneficial effects of cognitive remediation [70] and suggests potential benefits from specific exercises [71], while there is little evidence of efficacy regarding symptomatic pharmacological treatments in clinical trials, with mixed results showing clinical benefit in some individuals (amantidine [72]; fampiridine [73]; L-amphetamine [74]; lisdexamfetamine [75]; memantine [76]; rivastigmine [77]; donepezil [78]; ginkgo biloba [79]).

High-prevalence MS symptoms, such as fatigue, sleep, and mood disorders, have a negative impact on cognitive impairment [80,81,82,83,84,85,86]. In particular, depression and anxiety worsen memory, processing speed, and executive functions in MS patients, while sleep disorders and obstructive sleep apnoea are associated with impairment of visual and verbal memory, executive functions, attention, processing speed, and working memory [82,83,84,85,86]. It follows that good management of fatigue, mood, and sleep disorders may improve cognitive impairment in MS [87,88].

MS patients with significant CI have less engagement in social activities, lower employment rates, and an increased risk of developing psychiatric diseases [20], highlighting the clinical and socio-economic importance of cognitive involvement as a marker of disease severity [89].

The amount of data on CI derived from observational studies and RCTs is constantly growing. The available DMTs for MS treatment have important anti-inflammatory properties with less pronounced neuroprotective effects [27,28,29,30]. Although there are still insufficient data on their impact on CI, in this review we aimed to give an overview of the available data regarding the effect of oral DMTs on CI.

Considering the longer experience with this drug class, S1P modulators are covered by a significantly higher number of studies prospectively evaluating cognition in PwMS. Moreover, only S1P modulators have data on progressive MS, FTY on primary progressive MS, and SIP on secondary progressive MS, respectively [33,49,54].

PwMS treated with S1P modulators seem to have slight improvement or stability in cognitive test scores over time. Moreover, three different studies report a superiority of FTY compared to injectable DMTs [43,46,52]. No clear difference has been seen compared to other oral DMTs nor compared to NTZ or ATZ [45,56,57]. Among the S1P modulators, FTY certainly has the most evidence, since it is the drug that has been on the market the longest. The CI data for SIP and OZN are substantially derived from the EXPAND and SUNBEAM RCTs, respectively [33,39,49,50,54].

More than half of the papers on S1P modulators provide data on brain atrophy. The evidence would seem to show that in patients treated with S1P modulators, there is a reduction in BVL over time compared to placebo and IFN, while showing discordant results for GA and no difference compared to NTZ [45,46,56]. BVL also seems to correlate with cognitive performance in two of the included studies [53,55]. On the other hand, data on NFL are less abundant, mostly deriving from the INFORMS (FTY) and EXPAND (SIP) RCTs and a real-world study in which exposure to FTY and SIP would appear to be associated with a reduction in pNFL levels over time [54,57].

With regard to DMF, we retrieved data from the RCTs DEFINE and CONFIRM and six prospective studies. These studies would seem to indicate that patients treated with DMF show longitudinal improvement or stability in cognitive performance over time. However, there is a study where DMF and TFL (Category 3) were associated with significant worsening of cognitive performances vs. Category 1 (NTZ and ATZ) and Category 2 (FTY and CLAD) DMTs [56]. No significant differences were shown in the only other study analysed in this review against a DMT (FTY) [57]. Only one of the included studies also reported data on brain atrophy, which found a BVL of less than 0.4% (NEDA-4 annualised BVL cut-off) after one year of treatment [62]. The same applies to NFL, of which we have data from only one study vs. FTY where they were analysed at a single timepoint after 7 months from the start of treatment, showing higher values in the FTY group [57].

Concerning TFL, on the other hand, data are available from five prospective studies and a post hoc analysis of the RCT TEMSO and its extension study. These studies suggest mild improvement or stability in cognitive scores over time in PwMS treated with TFL. Data including an active DMT comparator come from two studies with a very low sample size, so there are insufficient data [45,56]. On brain atrophy, the data mostly come from the RCT TEMSO and its extension, where an inverse correlation between BVL and PASAT score was shown [63].

Data on CLAD, on the other hand, have only been analysed in one paper, with a small sample size (*n* = 13) [56]. In this paper, Glasmacher et al. grouped different DMTs together without distinguishing the individual DMTs in their analysis, thus not permitting the evaluation of each treatment on cognitive outcomes. So, no conclusions can be drawn to date on the effects of this CLAD on CI.

Although the data seem encouraging, there are several limitations in these studies. For example, the cognitive assessments used are very heterogeneous, comprising both comprehensive batteries such as Rao’s BRB-NT and more rapidly used tests such as the SDMT or PASAT. The PASAT and the SDMT were the most widely used tests, and although they are validated instruments in MS, they still have limitations. In fact, both PASAT and SDMT are focused on information processing speed, which despite being the most affected cognitive domain, is not the only one [3,19,20,21,22,23]. Other evaluation tools such as Rao’s BRB-NT were used in few papers, making it difficult to compare results across studies. Moreover, another factor to consider is that these assessment tools over time may be vulnerable to the practice effect that may overestimate performances [90,91].

Papers analysed in this review are a mixture of RCTs, observational prospective studies, and post hoc analyses. Study durations were not homogeneous, ranging from 6 to 120 months, generating potential variability in the results. In several studies, patients were evaluated longitudinally without a comparator group. That is likely to limit the ability to attribute changes in cognitive outcomes directly to the treatments under investigation. In addition, due to their design, observational studies and post hoc analyses are more susceptible to bias than RCTs. In fact, the most solid data, especially in the case of S1P modulators, come to us from RCTs and their extensions, where the patients had the longest follow-up. In these studies, the presence of a selection bias could be assumed, with better responding patients having the highest probability of continuing the study. On the other hand, the study durations (in most cases, lower than 24 months) can be inadequate to demonstrate a significant beneficial effect on the CI outcomes. As already mentioned, there are to date insufficient data about oral DMTs and their impact on BVL and sNFL. It is not possible at the present time to draw definitive conclusions on these outcomes.

In view of these considerations, we can state that cognitive disorders continue to be a large terrain that has not yet been fully explored and needs further analysis. The assessment of the cognitive sphere must unquestionably become part of the routine at MS centres, by carrying out annual assessments in all patients even in the early stages of the disease [69]. When facing an MS patient with cognitive disorders, a suitable strategy could be to flank DMT with the use of symptomatic therapies, cognitive remediation, and cognitive exercises.

## Figures and Tables

**Figure 1 bioengineering-10-00848-f001:**
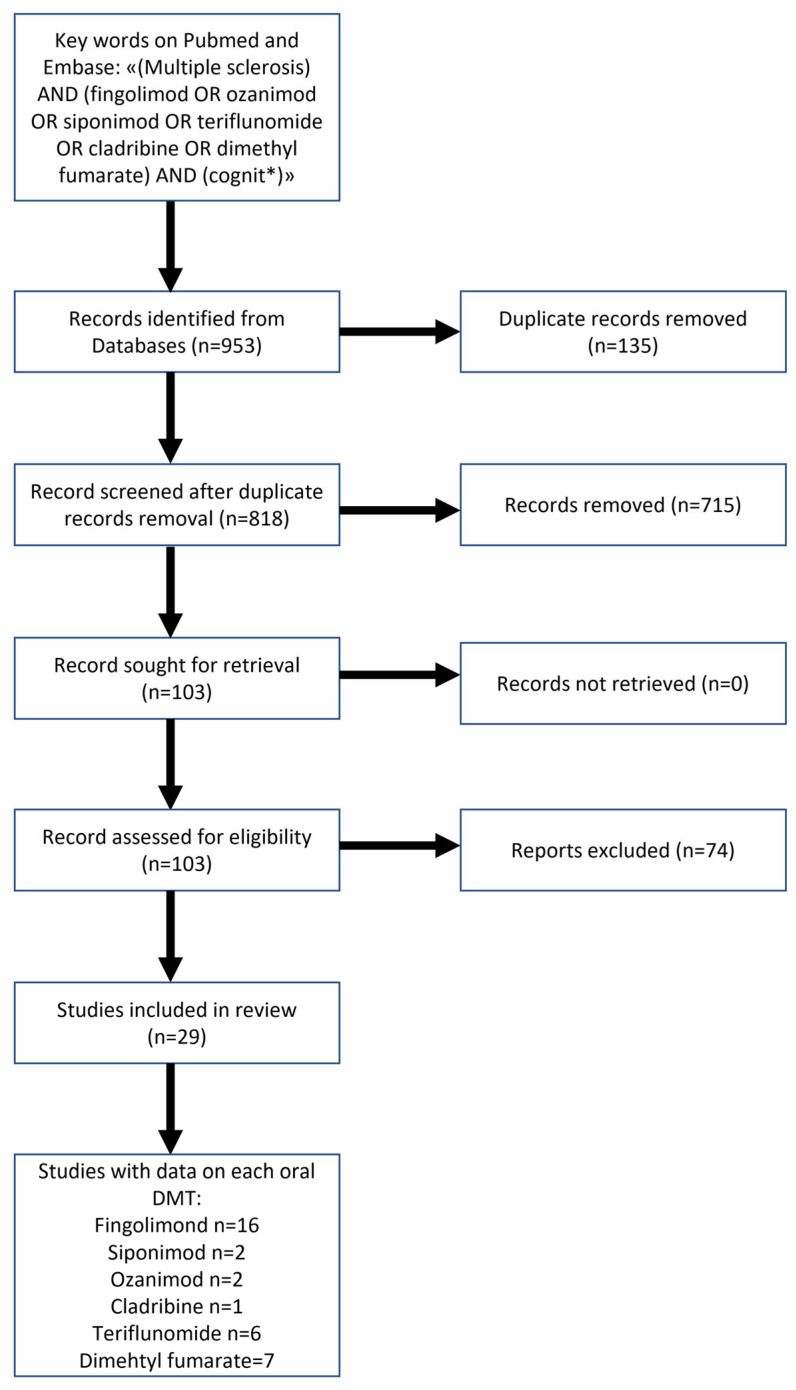
Study flow-chart.

**Table 1 bioengineering-10-00848-t001:** S1P Modulators.

Study	Type of Study	No. of Patients	Disease Form	Comparator (*n*)	Duration	Cognitive Outcome Measure	Results	Brain Atrophy Measure
Kappos et al., 2016 [40]	Post hoc analysis of 2 RCT FREEDOMS and FREEDOMS II (FTY)	783	RRMS	Placebo (773)	2 y	PASAT	PASAT improvements in FTY groups compare to placebo *p* < 0.0001.	37% higher BVL in placebo group.
Utz et al., 2016 [41]	Prospective (FTY)	22	RRMS	NTZ (11) IFN (7)	1 y	SPART/SPARTDR/PASAT 3/digit span forward/digit span backward/spatial span forward/spatial span backward/logical memory/go–no-go RT/go–no-go errors/divided attentions/visual search RT/visual search MT	No differences between groups. SDMT and PASAT 3 improved longitudinally in all groups.	
Comi et al., 2017 [42]	Multicentre randomised study (FTY)	97	RRMS	IFN1b (30)	18 m	Rao’s BRB-NT	RAO’s BRB-NT (0–18 months) improve significantly in both groups. No significant differences between groups.	PBVC significantly higher in IFN1b.
Cree et al., 2018 [43]	Phase 4 RCT PREFERMS (FTY)	433	RRMS	iDMTs (428)	1 y	SDMT	Increases were greater with FTY than with iDMTs at all assessments, but the between-group differences were not significant except at last assessment among patients taking the oral test.	In FTY group, PBVC and cortical GM volume reductions were significantly lower.
Petsas et al., 2019 [44]	Prospective (FTY)	32	RRMS	None	6 m	PASAT 2′/PASAT 3′	Improvement compared to baseline *p* = 0.016 (PASAT 3′) and *p* = 0.01 (PASAT 2′).	0.27% of BVL at the end of observation.
Comi et al., 2019 [39]	RCT phase 3 SUNBEAM (OZN)	447 (OZN 1 mg); 451 (OZN 0.5 mg)	RRMS	IFN beta 1a (448)	1 y	SDMT	Both OZN groups improved SDMT significantly compared to IFN.	Both OZN groups reduced BVL significantly compared to IFN.
Guevara et al., 2020 [45]	Prospective (FTY)	8	RRMS	TFL (4)/IFN (23)/GA (6)	2 y	SDMT	At 1 year and 2 year, 16% (7/45) and 20% (8/45) of patients had a SDMT decreased 4 points or more, respectively.	In all patients from baseline to 1.8 years WM volume, peripheral GM and whole brain volume decreased significantly.
Honce et al., 2020 [46]	Prospective (FTY)	44	RRMS	GA (43)	2 y	SDMT/PASAT 2′/CVLTII/BVMTR/COWAT/DKEFS	Significant improvement over time in both groups in most all cognitive assessments. No differences between groups except for DKEFS (better in FTY group).	No differences between the two groups in terms of annualised PBVC.
Preziosa et al., 2020 [47]	Prospective (FTY)	25	RRMS	NTZ (30)	2 y	Rao’s BRB-NT	Both groups improved at the end of study. No differences between groups.	Progressive atrophy during observation. No differences between groups.
Bhattacharyya et al., 2020 [48]	Prospective (FTY)	25	RRMS	None	2 y	PASAT/SDMT	Significant improvement in PASAT score. No changes in SDMT.	
Benedict et al., 2020 [49]	Secondary analysis of RCT phase 3 EXPAND (SIP)	1099	SPMS	Placebo	2 y	SDMT, PASAT, BVMTR	SDMT improved in SIP group compared to placebo. SIP-treated patients were at significantly lower risk for having a 4-point sustained decrease in SDMT score.	
Deluca et al., 2020 [50]	Post hoc analysis of SUNBEAN (OZN)	447 (OZN 1 mg); 451 (OZN 0.5 mg)	RRMS	IFN beta 1a (448)	1 y	SDMT	Both OZN groups improved SDMT significantly compared to IFN (*p* < 0.05).	
Ozakbas et al., 2021 [51]	Real world prospective (FTY)	356	RRMS	None	5 y	BICAMS	Significant improvement from baseline to year 5 on each BICAMS component.	
El Ayoubi et al., 2021 [52]	Prospective (FTY)	71	RRMS	Interferons (56)	1 y	SDMT/MoCA/total recall/delayed recall	SDMT improved longitudinally in both groups. MoCA and total recall worsened in interferon group and improved in FTY.	
Langdon et al., 2021 [53]	Post hoc analysis of 2 RCTs FREEDOMS and FREEDOMS II (FTY)	783	RRMS	Placebo (773)	10 y	PASAT 3	PASAT 3 improvements in FTY groups compared to placebo *p* < 0.0001.	Low EDSS and low BV correlates with better PASAT results.
Leppert et al., 2022 [54]	Post hoc analysis of RCTs INFORMS and EXPAND (FTY and SIP)	303 (FTY) 1272 (SIP)	PPMS and SPMS	Placebo (1033)	2 and 3 y	PASAT/SDMT	High pNFL correlates with CI. FTY and SIP reduced pNFL *p* < 0.05 and *p* < 0.01	High pNFL correlates with BVL.
Cree et al., 2022 [33]	Secondary analysis of RCT phase 3 EXPAND (SIP) and up to >5 y extension (ongoing)	593	SPMS	Placebo until y 2, then SIP.	5 y	SDMT	Six-month CCW risk significantly lower in continuous SIP vs. placebo-SIP.	Continuous SIP has significantly lower PBVC and thalamic volume reduction.
Conway et al., 2022 [55]	Prospective (FTY)	15	RRMS	Healthy controls (5)	1 y	BVMTR/ SDMT/ DKEFS/SRT	Significant improvement over time in SDMT, BVMTR (total recall), DKEFS (number letter switching) in RRMS group. No differences between groups.	Thalamic volume and cortical thickness are significant predictors of CI.
Glasmacher et al., 2022 [56]	Prospective (DMTS category 2 FTY and CLAD)	10 (FTY) 13 (CLAD)	RRMS	Category 1 ATZ (25) and NTZ/Category 3 DMF (148) and TFL (11)/No DMTS (117)	1 y	PASAT	Category 2 associated with significant improvement; Category 1 not associated with improvement or worsening; Category 3 associated with significant worsening.	
Hersh et al., 2022 [57]	Real-world prospective (FTY)	541	RRMS	DMF (632)	1 y	PST	At baseline, FTY group had higher PST score. No significant changes over time between FTY and DMF groups. sNFL data available for some patients at single timepoint (7 months): higher levels in FTY group.	

ATZ Alemtuzumab; BICAMS Brief International Cognitive Assessment for Multiple Sclerosis; BRB-NT Brief Repeatable Battery of Neuropsychological Tests; BVL Brain Volume Loss; BVMTR Brief Visuospatial Memory Test Revised; CCW Confirmed Cognitive Worsening; CLAD Cladribine; COWAT Controlled Oral Word Association; CVLTII California Verbal Learning Test II; DKEFS Delis-Kaplan Executive Function System test; DMF Dimethyl Fumarate; DMTs Disease-Modifying Therapies; EDSS Expanded Disability Status Scale; FTY Fingolimod; GA Glatiramer Acetate; GM Grey Matter; iDMTs injectable DMTs; IFN Interferon beta; MoCA Montreal Cognitive Assessment; MSPS Multiple Sclerosis Performance Scale; MT Movement Time; NTZ Natalizumab; OZN Ozanimod; PASAT Paced Auditory Serial Addition Task; PBVC Percent Brain Volume Change; pNFL plasma Neurofilament-Light chain; PST Processing Speed Test; RCT Randomised Clinical Trial; RRMS Relapsing Remitting Multiple Sclerosis; RT Reaction Time; SDMT Symbol Digit Modality test; SIP Siponimod; sNFL serum Neurofilament-Light Chain; SPART Spatial Recall Test; SPARTDR Delayed Recall of the Spatial Recall Test; SRT Selective Reminding Test; TFL Teriflunomide; WM White Matter.

**Table 2 bioengineering-10-00848-t002:** Dimethyl fumarate (DMF).

Study	Type of Study	No. of Patients	Disease Form	Comparator (*n*)	Duration	Cognitive Outcome Measure	Results	Brain atrophy Measure
Giovannoni et al., 2016 [58]	Integrated analysis of phase 3 RCTs (CONFIRM and DEFINE)	769	RRMS	Placebo (771)	96 w	PASAT 3′ (mean change)	Mean change in PASAT 3′ z-scores was 0.178 for DMF and 0.123 for placebo (*p* = 0.0016).	
Al Iedani et al., 2018 [59]	Prospective	20	RRMS	Healthy controls	2 y	SDMT	SDMT score stable at year 2 in MS patients.	No substantial change was observed in the average annualised rate of brain volume loss between 1st and 2nd year of treatment with DMF.
Montes Diaz et al., 2018 [60]	Prospective	16	RRMS	None	1 y	PASAT	Significant improvement after 3 months.	
Amato et al., 2020 [61]	Prospective	156	RRMS	None	2 y	Stroop test and BRB-NT	CI in 22.6% patients at baseline, in 27.2% at 1 year, and 9.7% at 2 years. Compared to year 1, 37.2% improved, 10.7% worsened, and 52.1% remained unchanged at 2 years.	
Hersh et al., 2022 [57]	Real-world prospective	632	RRMS	FTY (541)	1 y	PST	At baseline, FTY group had higher PST score. No significant changes over time between FTY and DMF groups. sNFL data available for some patients: higher baseline levels in FTY group.	
Piervincenzi et al., 2022 [62]	Prospective	27	RRMS	None	1 y	PASAT 3′, SDMT	PASAT improvement at month 12 (*p* = 0.022). SDMT unchanged.	−0.12% BVL at month 6; −0.24% at month 12.
Glasmacher et al., 2022 [56]	Prospective (DMTS category 3 DMF and TFL)	148 (DMF) 11 (TFL)	RRMS	Category 1 ATZ (25) and NTZ/Category 2 FTY (10) and CLAD (13)/No DMTS (117)	1 y	PASAT	Category 2 associated with significant improvement; Category 1 not associated with improvement or worsening. Category 3 associated with significant worsening.	

ATZ Alemtuzumab; BRB-NT Brief Repeatable Battery of Neuropsychological Tests; BVL Brain Volume Loss; CI Cognitive Impairment; CLAD Cladribine; DMF Dimethyl Fumarate; DMTs Disease-Modifying Therapies; FTY Fingolimod; GA Glatiramer Acetate; MS Multiple Sclerosis; NTZ Natalizumab; PASAT Paced Auditory Serial Addition Task; PST Processing Speed Test; RCT Randomised Clinical Trial; RRMS Relapsing Remitting Multiple Sclerosis; SDMT Symbol Digit Modality test; sNFL Serum Neurofilament-Light Chain; TFL Teriflunomide.

**Table 3 bioengineering-10-00848-t003:** Teriflunomide.

Study	Type of Study	No. of Patients	Disease Form	Comparator (*n*)	Duration	Cognitive Outcome Measure	Results	Brain atrophy Measure
Coyle et al., 2018 [64]	Open-label multicentre Phase IV Study (Teri PRO)	594	RRMS	None	2 y	SDMT	SDMT results remained stable at week 48. CI reported by patients on MSPS remained stable.	
Guevara et al., 2020 [45]	Prospective	4	RRMS	FTY (8)/IFN (23)/GA (6)	2 y	SDMT	At 1 year and 2 year, 16% (7/45) and 20% (8/45) of patients had a SDMT decreased 4 points or more, respectively.	In all patients from baseline to 1.8 years after baseline, WM volume, peripheral GM, and whole brain volume decreased significantly.
Corallo et al., 2021 [65]	Prospective	30	RRMS	Healthy controls (30)	1 y	Rao’s BRB-NT	No significant changes from baseline in each cognitive test of BRB-NT.	Mild increase in GM volume. No WM volume loss.
Bencsik et al., 2022 [66]	Real-world study (TERI REAL) analysis	146	RRMS	None	2 y	BICAMS	Mild improvement at month 12 and 24 *p* < 0.05 of each BICAMS component.	
Sprenger et al., 2022 [63]	Post hoc analysis of RCT TEMSO and its extension	235	RRMS	Placebo	252 w	PASAT 3′	PASAT 3′ Z-scores increased through week 252 post-TFL initiation. Patients who received 14 mg TFL in the core study and extension had higher PASAT 3′ Z-scores through week 252 compared with those who switched at the extension.	Lower BVL on TFL treatment correlated with higher PASAT 3′ scores.
Glasmacher et al., 2022 [56]	Prospective (DMTs Category 3 DMF and TFL)	148 (DMF) 11 (TFL)	RRMS	Category 1 ATZ (25) and NTZ/Category 2 FTY (10) and CLAD (13)/no DMTs (117)	1 y	PASAT	Category 2 associated with significant improvement; Category 1 not associated with improvement or worsening; Category 3 associated with significant worsening.	

ATZ Alemtuzumab; BICAMS Brief International Cognitive Assessment for Multiple Sclerosis; BRB-NT Brief Repeatable Battery of Neuropsychological Tests; BVL Brain Volume Loss; CLAD Cladribine; DMF Dimethyl Fumarate; DMTs Disease-Modifying Therapies; FTY Fingolimod; GA Glatiramer Acetate; GM Grey Matter; IFN Interferon beta; MSPS Multiple Sclerosis Performance Scale; NTZ Natalizumab; PASAT Paced Auditory Serial Addition Task; RCT Randomised Clinical Trial; RRMS Relapsing Remitting Multiple Sclerosis; SDMT Symbol Digit Modality test; TFL Teriflunomide; WM White Matter.

## Data Availability

The datasets generated during and/or analyzed during the current study are available from the corresponding author on reasonable request.

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
