# Peer review of "Current Status of Oral Disease-Modifying Treatment Effects on Cognitive Outcomes in Multiple Sclerosis: A Scoping Review"

_bioengineering, 2023, doi:10.3390/bioengineering10070848_

Round 1

Reviewer 1 Report

Dear authors,

the article is interesting and well crafted. Still, some adjustments are advised in my opinion.

1) Thorough English check is needed;

2)I would remodulate concepts organisation in the "Introduction" section, as they do not flow organically, explaining more clearly on which evidence (if present) shown by present scientific literature you based the idea of your review. 

3)In M&M you wrote "Out of the 818 detected articles, 715 were excluded for ineligibility (not concern-129 ing cognition; animal studies; not concerning oral DMTs; reviews; case reports; book chap-130 ters; editorials). 103 articles were then analysed, of which 75 were excluded because they 131 did not provide sufficient data on cognitive assessments. Therefore 29 studies were in-132 cluded in this review (Figure 1).". These are results. You should have provided inclusion and exclusion criteria in M&M and then present your results in the subsequent section. 

4)Discussion section is too laced.

After these improvement, I believe the article will be suitable for pubblication.

Kind regards

English check is required

Author Response

REV1

Dear reviewer,

We revised our manuscript following your recommendations as it follows:

1) Thorough English check is needed

Thank you for your suggestions. We made an English check and the consequent corrections.

2)I would re-modulate concept organization in the "Introduction" section, as they do not flow organically, explaining more clearly on which evidence (if present) shown by present scientific literature you based the idea of your review.

4)Discussion section is too laced.

Thank you for your suggestions. To be more clear and straightforward, we modulated both the introduction and the discussion sections to better specify the objectives of our review and to explain the importance of cognitive impairment in MS people and the limit of the available literature.

3)In M&M you wrote "Out of the 818 detected articles, 715 were excluded for ineligibility (not concern-129 ing cognition; animal studies; not concerning oral DMTs; reviews; case reports; book chap-130 ters; editorials). 103 articles were then analysed, of which 75 were excluded because they 131 did not provide sufficient data on cognitive assessments. Therefore 29 studies were in-132 cluded in this review (Figure 1).". These are results. You should have provided inclusion and exclusion criteria in M&M and then present your results in the subsequent section.

Thank you for your suggestions. In line 125 we added the sentence “Exclusion criteria were the following: papers not concerning cognition; animal studies; papers not concerning oral DMTs; reviews; case reports; book chapters; editorials; cross-sectional studies.”. At line 131 we removed the period “The keywords used for the search were (Multiple sclerosis) AND (fingolimod OR ozanimod OR siponimod OR teriflunomide OR cladribine OR dimethyl fumarate) AND (cognit*). Out of the 818 detected articles, 715 were excluded for ineligibility (not concerning cognition; animal studies; not concerning oral DMTs; reviews; case reports; book chapters; or editorials). 103 articles were then analyzed, of which 75 were excluded because they did not provide sufficient data on cognitive assessments. Therefore 29 studies were included in this review (Figure 1).”. At line 139 in the “Results” section we inserted the period “The keywords used for the search were (Multiple sclerosis) AND (fingolimod OR ozanimod OR siponimod OR teriflunomide OR cladribine OR dimethyl fumarate) AND (cognit*). Out of the 818 detected articles, 715 were excluded for ineligibility.103 articles were then analyzed, of which 75 were excluded because they did not provide sufficient data on cognitive assessments. Therefore 29 studies were included in this review (Figure 1).”

Reviewer 2 Report

The review provides a detailed analysis of a variety of oral disease-modifying therapies (DMTs) and their impact on cognitive impairment in patients with multiple sclerosis (MS). By comparing different DMTs, the review gives a broad perspective on available treatments and their potential cognitive benefits. However, there were several flaws to be clarified.

1. The studies reviewed included a mixture of randomized controlled trials (RCTs), observational prospective studies, and post hoc analyses. The latter two designs are more prone to bias than RCTs, which could affect the accuracy of the conclusions drawn.

2. The duration of studies ranged from 12 to 58 months, which can introduce variability in the results.

3. In some studies, patients were evaluated longitudinally without a comparator or control group, which could limit the ability to attribute changes in cognitive outcomes directly to the treatments under investigation.

4. Some studies grouped different DMTs together without separating the individual treatments in their analyses, which makes it difficult to determine the effect of each treatment individually.

5. Different cognitive assessments were used across studies. Some cognitive assessments were used in only one or two studies, making it difficult to compare these results across studies or drugs.

6. In the study with Cladribine, drugs were grouped into categories, which could introduce potential biases depending on how these categories were determined.

7. Particularly for Cladribine, Dimethyl Fumarate, and outcomes such as brain volume loss (BVL) and neurofilament light chain (NFL), there was limited available data. This makes it challenging to draw firm conclusions about these drugs and their impact on cognitive impairment.

The language used in the text is generally well-written and clear. It uses standard scientific language, maintains professional tone, and uses jargon appropriate for the audience (likely researchers or practitioners in neuroscience, medicine, or related fields).

However, there are a few minor areas that could potentially be improved:

1. Readability: Some sentences are quite long, which can make them difficult to follow. For example, consider splitting the sentence starting with "About DMF, we retrieved data from the RCTs DEFINE and CONFIRM and six prospective studies..." into two for better readability.

2. Consistent Terminology: The text alternates between acronyms and full terms for the same concepts (e.g., "teriflunomide" and "TRF", "cognitive impairment" and "CI"). Consistently using either the full term or the acronym throughout the document might improve clarity and consistency.

3. Specific References: Occasionally, the text uses phrases like "one study" or "another study" without providing a specific citation immediately after. It would be beneficial to cite the specific study being referenced directly after these phrases to maintain clarity.

4. Subject-Verb Agreement: There are a few instances where the subject and verb do not agree in number. For example, the sentence "PwMS treated with S1P modulators seems to have a slight improvement..." should be corrected to "PwMS treated with S1P modulators seem to have a slight improvement...".

5. Punctuation: There are some issues with punctuation, including unnecessary spaces before punctuation marks, especially before commas and brackets. This might be due to formatting or typing errors.

6. Correct Use of Prepositions: There are a few instances where prepositions may not have been used correctly. For instance, "data on Cladribine, on the other hand, have only been analysed in one paper," could be more clearly written as "data on Cladribine, on the other hand, have only been analyzed in one paper."

Author Response

REV 2

Dear reviewer,

We revised our manuscript following your recommendations as follows.

1.The studies reviewed included a mixture of randomized controlled trials (RCTs), observational prospective studies, and post hoc analyses. The latter two designs are more prone to bias than RCTs, which could affect the accuracy of the conclusions drawn.

Thank you for your suggestion. To better explain this concept in the “Discussion” section we added in line 330 “Papers analyzed in this review are a mixture of RCTs, observational prospective studies and posthoc analysis.” and in line 335 “Due to their design, observational studies and posthoc analyses are more susceptible to bias than RCTs”.

2.The duration of studies ranged from 12 to 58 months, which can introduce variability in the results.

Thank you for your suggestion. We added at line 331 the sentence “Study durations were not homogeneous ranging from 6 to 120 months, generating potential variability in the results.”

3.In some studies, patients were evaluated longitudinally without a comparator or control group, which could limit the ability to attribute changes in cognitive outcomes directly to the treatments under investigation.

Thank you for your suggestion. We added at line 332 the period “In several studies, patients were evaluated longitudinally without a comparator group [80-82,85,86,91-93]. That is likely to limit the ability to attribute changes in cognitive outcomes directly to the treatments under investigation.

4.Some studies grouped different DMTs together without separating the individual treatments in their analyses, which makes it difficult to determine the effect of each treatment individually.

Thank you for your suggestion. We added at line 316 the period “In this paper [74] Glasmacher et al. grouped different DMTs together without separating the individual treatments in their analyses, which makes it difficult to determine the effect of each treatment individually.”

5.Different cognitive assessments were used across studies. Some cognitive assessments were used in only one or two studies, making it difficult to compare these results across studies or drugs.

Thank you for your suggestion. To better express this concept, we added at line 326 the sentence “Other evaluation tools such as Rao's BRB-NT were used in few papers, making it difficult to compare results across studies.”

6.In the study with Cladribine, drugs were grouped into categories, which could introduce potential biases depending on how these categories were determined.

Thank you for your suggestion. We added at line 316 the period “In this paper [74] Glasmacher et al. grouped different DMTs together without separating the individual treatments in their analyses, which makes it difficult to determine the effect of each treatment individually.”

7.Particularly for Cladribine, Dimethyl Fumarate, and Teriflunomide outcomes such as brain volume loss (BVL) and neurofilament light chain (NFL), there was limited available data. This makes it challenging to draw firm conclusions about these drugs and their impact on cognitive impairment.

Thank you for your suggestion. We added at line 342 the period “As already mentioned above, concerning the impact on BVL and sNFL there are to date insufficient data about oral DMTs. It is not possible at present to draw definitive conclusions on these outcomes.”

Comments on the Quality of English Language

The language used in the text is generally well-written and clear. It uses standard scientific language, maintains professional tone, and uses jargon appropriate for the audience (likely researchers or practitioners in neuroscience, medicine, or related fields).

However, there are a few minor areas that could potentially be improved:

  1. Readability: Some sentences are quite long, which can make them difficult to follow. For example, consider splitting the sentence starting with "About DMF, we retrieved data from the RCTs DEFINE and CONFIRM and six prospective studies..." into two for better readability.

To enhance the readability of the manuscript, we changed some sentences to shorter and clearer ones.

  1. Consistent Terminology: The text alternates between acronyms and full terms for the same concepts (e.g., "teriflunomide" and "TRF", "cognitive impairment" and "CI"). Consistently using either the full term or the acronym throughout the document might improve clarity and consistency.

We modified the text consistently using acronyms.

  1. Specific References: Occasionally, the text uses phrases like "one study" or "another study" without providing a specific citation immediately after. It would be beneficial to cite the specific study being referenced directly after these phrases to maintain clarity.

We modified the text providing references right after the citations.

  1. Subject-Verb Agreement: There are a few instances where the subject and verb do not agree in number. For example, the sentence "PwMS treated with S1P modulators seems to have a slight improvement..." should be corrected to "PwMS treated with S1P modulators seem to have a slight improvement...".

We corrected all subject-verb agreement issues throughout the text.

  1. Punctuation: There are some issues with punctuation, including unnecessary spaces before punctuation marks, especially before commas and brackets. This might be due to formatting or typing errors.

We corrected all punctuation issues throughout the text.

  1. Correct Use of Prepositions: There are a few instances where prepositions may not have been used correctly. For instance, "data on Cladribine, on the other hand, have only been analysed in one paper," could be more clearly written as "data on Cladribine, on the other hand, have only been analyzed in one paper.".

We checked and corrected all wrong use of prepositions throughout the text.

Round 2

Reviewer 1 Report

Dear Authors,

the article is well crafted and it is suitable for publication in my opinion. Kind regards.

Minor check needed

Reviewer 2 Report

The authors have thoroughly addressed all the concerns and criticisms raised by the reviewers. They have carefully considered the feedback provided and made the necessary revisions and improvements to their work.